# LITE TRANSFORMER WITH LONG-SHORT RANGE ATTENTION

**Zhanghao Wu**[*1,2]    **Zhijian Liu**[*1]    **Ji Lin**[1]    **Yujun Lin**[1]    **Song Han**[1]
[1]Massachusetts Institute of Technology    [2]Shanghai Jiao Tong University
{zhwu, zhijian, songhan}@mit.edu

## ABSTRACT

Transformer has become ubiquitous in natural language processing (*e.g.*, machine translation, question answering); however, it requires enormous amount of computations to achieve high performance, which makes it not suitable for mobile applications that are tightly constrained by the hardware resources and battery. In this paper, we present an efficient mobile NLP architecture, *Lite Transformer* to facilitate deploying mobile NLP applications on edge devices. The key primitive is the *Long-Short Range Attention* (LSRA), where one group of heads specializes in the **local** context modeling (by convolution) while another group specializes in the **long-distance** relationship modeling (by attention). Such specialization brings consistent improvement over the vanilla transformer on three well-established language tasks: machine translation, abstractive summarization, and language modeling. Under constrained resources (500M/100M MACs), Lite Transformer outperforms transformer on WMT'14 English-French by 1.2/1.7 BLEU, respectively. Lite Transformer reduces the computation of transformer base model by 2.5× with 0.3 BLEU score degradation. Combining with pruning and quantization, we further compressed the model size of Lite Transformer by **18.2×**. For language modeling, Lite Transformer achieves 1.8 lower perplexity than the transformer at around 500M MACs. Notably, Lite Transformer outperforms the AutoML-based Evolved Transformer by 0.5 higher BLEU for the mobile NLP setting without the costly architecture search that requires more than 250 GPU **years**. Code has been made available at https://github.com/mit-han-lab/lite-transformer.

## 1 INTRODUCTION

Transformer (Vaswani et al., 2017) is widely used in natural language processing due to its high training efficiency and superior capability in capturing long-distance dependencies. Building on top of them, modern state-of-the-art models, such as BERT (Devlin et al., 2019), are able to learn powerful language representations from unlabeled text and even surpass the human performance on the challenging question answering task.

However, the good performance comes at a high computational cost. For example, a single transformer model requires more than 10G Mult-Adds in order to translate a sentence of only 30 words. Such extremely high computational resources requirement is beyond the capabilities of many edge devices such as smartphones and IoTs. Therefore, it is of great importance to design efficient and fast transformer architecture specialized for real-time NLP applications on the edge. Automatic neural architecture search (Zoph & Le, 2017; So et al., 2019) is a choice for high accuracy model design, but the massive search cost (GPU hours and $CO_2$ emission) raises severe environmental concerns (Strubell et al., 2019), shown in Figure 1b.

In this paper, we focus on the efficient inference for mobile devices, where the total number of Mult-Adds is constrained below 500M. A straightforward way to reduce the computation of the transformer is to shrink the embedding size directly. Although it can effectively reduce both model size and computation, it also weakens the model capacity capturing the long and short distance relationship at the same time. To this end, we systematically studied the computation breakdown of the transformer

---

* indicates equal contributions.

and observed that the computation (Mult-Adds) is dominated by the feed-forward network (FFN). We discovered that the prevailing bottleneck-structured transformer block is not efficient. We then present a novel Long-Short Range Attention (LSRA) primitive. LSRA trades off the computation in FFN for wider attention layers. It stretches the bottleneck to introduce more dependency capturing capability for the attention layer, and then shrink the embedding size to reduce the total computation amount while maintaining the same performance. Instead of having one module for "general" information, LSRA dedicates *specialized* heads to model long and short distance contexts. Inspired by Wu et al. (2019b), LSRA introduces convolution in a parallel branch to capture **local** dependencies so that the attention branch can focus on **global** context capture. By stacking this primitive, we build Lite Transformer for mobile NLP applications.

Extensive experiments demonstrate that our Lite Transformer model offers significant improvements over the transformer on three language tasks: machine translation, abstractive summarization, and language modeling. For machine translation, on IWSLT 2014 German-English, it outperforms the transformer by 3.1 BLEU under 100M Mult-Adds; on WMT 2014 English-German, it surpasses the transformer by 0.4 BLEU under 500M Mult-Adds and 1.2 BLEU under 100M Mult-Adds; on WMT 2014 English-French, it also achieves consistent improvements over the transformer: 1.2 BLEU under 500M Mult-Adds and 1.7 BLEU under 100M Mult-Adds. Further, combined with general model compression techniques (pruning and quantization), our Lite Transformer can achieve $18.2\times$ model size compression. For the summarization task, on CNN-DailyMail, it reduces the computation of the transformer base model by $2.4\times$. For language modeling, it achieves 1.8 lower perplexity than the transformer around 500M Mult-Adds.

Guided by our design insights, our manually-designed Lite Transformer achieves 0.5 higher BLEU than the AutoML-based Evolved Transformer (So et al., 2019), which requires more than 250 GPU years to search, emitting as much carbon as five cars in their lifetimes (see Figure 1b). It indicates that AutoML is not a panacea: careful analysis and design insights (*i.e.*, removing the bottleneck, specialized heads) can effectively prune the search space and improve the sample efficiency.

The contribution of this paper has four aspects:

1. We systematically analyze the commonly used computation bottleneck structure in modern neural networks and find that the bottleneck design is not optimal for 1-D attention if using FLOPs as figure of merit.

2. We propose a specialized multi-branch feature extractor, Long-Short Range Attention (LSRA), as the basic building block of our transformer, where convolution helps capture the local context and attention concentrates on global context.

3. We build Lite Transformer based on our LSRA. Under mobile computation resource constraints (500M Mult-Adds), our Lite Transformer demonstrates coherent improvement on three widely used machine translation datasets. With extra experiments on other tasks, Lite Transformer is efficient for multiple language applications.

4. Even compared to AutoML-searched Evolved Transformer, our Lite Transformer offers 0.5 higher BLEU score on WMT En-De dataset under the mobile setting, saving the design cost by $20000\times$ in $CO_2$ emissions. It alerts us to rethink the practicality of AutoML in terms of design cost and "green AI".

## 2  RELATED WORK

**RNNs and CNNs.**  Recurrent neural networks (RNNs) have prevailed various sequence modeling tasks for a long time (Sutskever et al., 2014; Luong et al., 2015; Bahdanau et al., 2015; Wu et al., 2016). However, RNNs are not easy to parallelize across the sequence due to its temporal dependency. Recently, some work has demonstrated that RNN is not an essential component to achieve state-of-the-art performance. For instance, researchers have proposed highly-efficient convolution-based models (Kalchbrenner et al., 2016; Gehring et al., 2017; Kaiser et al., 2018; Wu et al., 2019b). Convolution is an ideal primitive to model the local context information; however, it lacks the ability to capture the long-distance relationship, which is critical in many sequence modeling tasks.

**Transformers.**  As an alternative, attention is able to capture global-context information by pairwise correlation. Transformer (Vaswani et al., 2017) has demonstrated that it is possible to stack the

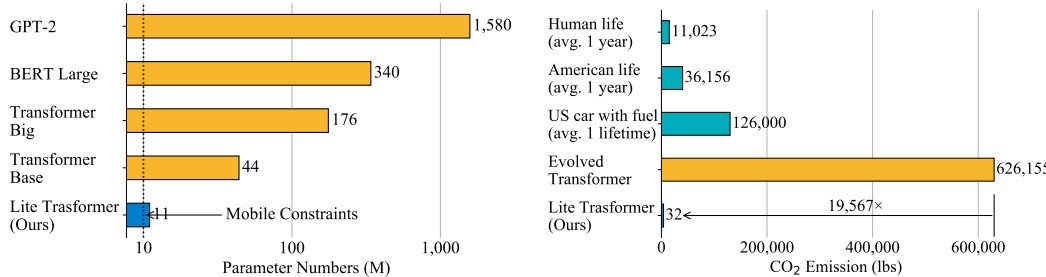

(a) Parameter numbers of modern NLP models.  (b) The design cost measured in $CO_2$ emission (lbs).

Figure 1: Left: the size of recent NLP models grows rapidly and exceeds the mobile constraints to a large extent. Right: the search cost of AutoML-based NLP model is prohibitive, which emits carbon dioxide nearly $5\times$ the average lifetime emissions of the car.

self-attentions to achieve state-of-the-art performance. Recently, there have been a lot of variants to the transformer (Ahmed et al., 2017; Ott et al., 2018; Chen et al., 2018; Paulus et al., 2018; Shaw et al., 2018; Sukhbaatar et al., 2019a;b; Child et al., 2019). Among them, Ott et al. (2018) proposed to scale up the batch size; Shaw et al. (2018) leverages the relative position representations; Ahmed et al. (2017) introduces the weighted multi-head attention; Sukhbaatar et al. (2019a) applies adaptive masks for long-range information on character-level language modeling with very long sequences. All these attempts are orthogonal to our work, as their methods can also be applied in our architecture.

**Automated Model Design.**   Due to the vast architecture design space, automating the design with neural architecture search (NAS) becomes popular (Zoph & Le, 2017; Zoph et al., 2018; Pham et al., 2018; Cai et al., 2019a). To make the design efficient, integrating the hardware resource constraints into the optimization loop begins to emerge, such as MnasNet (Tan et al., 2019), ProxylessNAS (Cai et al., 2019b) and FBNet (Wu et al., 2019a). In the NLP community, the evolved transformer (So et al., 2019) adopts the neural architecture search (Zoph & Le, 2017) to design basic blocks and finds a better #parameter-BLEU trade-off for the transformer. However, AutoML-based model designs require significant amount of GPU hours to find the 'best' model, which is not affordable for most researchers.

**Model Acceleration.**   Apart from designing efficient models directly (Liu et al., 2019b; Li et al., 2020), another approach to achieve efficient inference is to compress and accelerate the existing large models. For instance, some have proposed to prune the separate neurons (Han et al., 2015b; 2016) or the entire channels (He et al., 2017; Liu et al., 2017; He et al., 2018); others have proposed to quantize the network (Courbariaux et al., 2016; Zhu et al., 2017; Krishnamoorthi, 2018; Wang et al., 2019) to accelerate the model inference. Recently, AutoML has also been used to automate the model compression and acceleration (He et al., 2018; Yang et al., 2018; Wang et al., 2019; Liu et al., 2019a). All these techniques are compressing existing models and are therefore orthogonal to our approach. We aim to explore how to make use of the domain knowledge to design an efficient architecture from the beginning, rather than compressing an existing model.

## 3   IS BOTTLENECK EFFECTIVE FOR 1-D ATTENTION?

The attention mechanism has been widely used in various applications, including 1-D (language processing (Vaswani et al., 2017)), 2-D (image recognition), and 3-D (video recognition (Wang et al., 2018)). It computes pairwise dot-product between all the input elements to model both short-term and long-term relationships. Despite its effectiveness, the operation introduces massive computation. Assume the number of elements (*e.g.*, length of tokens in language processing, number of pixels in image, *etc.*) fed to attention layer is $N$, and the dimension of features (*i.e.*, channels) is $d$, the computation needed for the dot-product is $N^2d$. For images and videos, $N$ is usually very large. For example, the intermediate feature map in a video network (Wang et al., 2018) has 16 frames, each with $112\times112$ resolution, leading to $N = 2 \times 10^5$. The computation of convolution and fully-connected layers grows linearly w.r.t. $N$, while the computation of attention layers grows quadratically w.r.t. $N$. The computation of attention module will soon overwhelm with a large $N$.

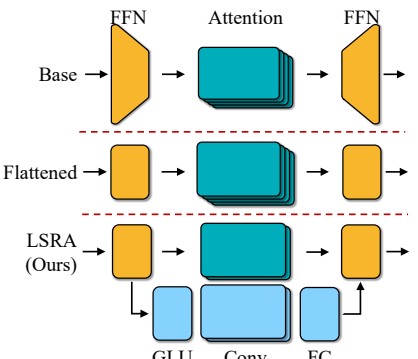 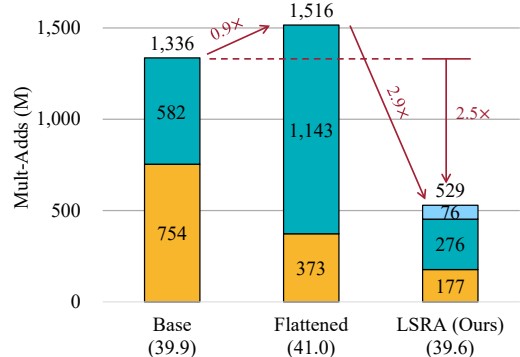

Figure 2: Flattening the bottleneck of transformer blocks increases the proportion of the attention versus the FFN, which is good for further optimization for attention in our LSRA.

To address the dilemma, a common practice is first to reduce the number of channels $d$ using a linear projection layer before applying attention and increase the dimension afterwards (as shown in Figure 2). In the original design of transformer (Vaswani et al., 2017), the channel dimension in the attention module is $4\times$ smaller than that in the FFN layer. Similarly, in the non-local video network (Wang et al., 2018), the channel number is first reduced by half before applying the non-local attention module. This practice saves the computation by $16\times$ or $4\times$. Nevertheless, it also *decreases* the contexts capture ability of attention layers with a smaller feature dimension. The situation could be even worse for language processing, as attention is the major module for contexts capture (unlike images and videos where convolutions conduct the major information capture).

For tasks like translation, the length of the input sequence $N$ tends to be small, which is around 20-30 in common cases. A transformer block consists of an attention (or two for decoder), followed by a feed-forward network (FFN). For the attention layer, the Mult-Adds would be $\mathcal{O}(4Nd^2 + N^2d)$; for FFN, the Mult-Adds is $\mathcal{O}(2 \times 4Nd^2)$. Given a small $N$, it is doubtful if the bottleneck design is a good trade-off between computation and accuracy on 1D attention. To verify the idea, we first profile the computation breakdown in the transformer in Figure 2. Surprisingly, for the original transformer (denoted as 'Base' in the figure), the FFN layer actually consumes much of the computation. This is not desirable since FFN itself cannot perform any contexts captures. In conclusion, due to the small $N$, the bottleneck design cannot significantly reduce the computation in 1D attention, while the limited benefit for computation reduction is further compromised by the large FFN layer. It also harms the capacity of attention layer due to the smaller dimension, which is the major contexts capture unit in the transformer.

Therefore, we argue that the bottleneck design is not optimal for 1-D attention. We instead design a 'flattened' version of the transform block that does not reduce and increase the channel dimension. With the new design, the attention part now takes up the major computation in the flattened transformer model in Figure 2, leaving a larger space for further optimization. We also test the performance change of such modification on WMT'14 En-Fr dataset. We can achieve comparable performance at a slightly larger computation, which can be easily reduced with further optimization that is discussed in the next section.

## 4 LONG-SHORT RANGE ATTENTION (LSRA)

Researchers have tried to understand the contexts captured by attention. Kovaleva et al. (2019) and Clark et al. (2020) visualized the attention weights from different layers in BERT. As shown in Figure 3b, the weights $w$ illustrate the relationships between the words from the source sentence and the target sentence (the same for self-attention). With a larger weight $w_{ij}$ (darker color), the $i$-th word in the source sentence pays more attention to the $j$-th word in the target sentence. And the attention maps typically have strong patterns: sparse and diagonal. They represent the relationships between some particular words: the sparse for the long-term information, and the diagonal for the correlation in small neighborhoods. We denote the former as "global" relationships and the latter as "local".

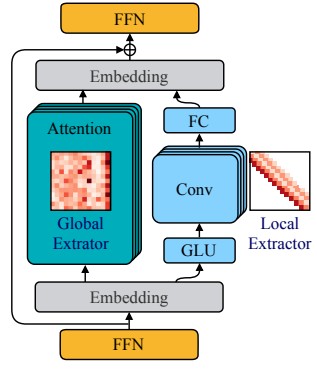

(a) Lite Transformer block

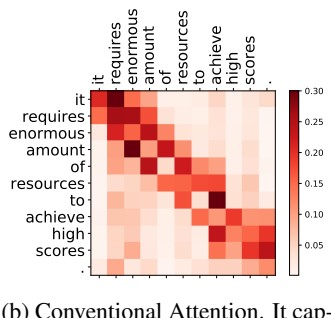

(b) Conventional Attention. It captures local information on the diagonal and global context as sparse points. (Redundant)

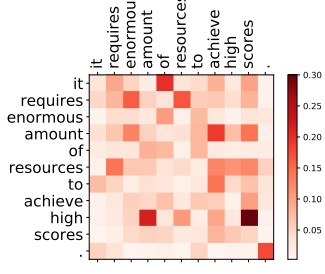

(c) Attention in LSRA. It is specialized for long-term relationships, indicated as points away from the diagonal. (Efficient)

Figure 3: Lite Transformer architecture (a) and the visualization of attention weights. Conventional attention (b) puts too much emphasis on local relationship modeling (see the diagonal structure). We specialize the local feature extraction by a convolutional branch which efficiently models the locality so that the attention branch can specialize in global feature extraction (c). More visualizations are available in Figure A1.

For a translation task, the attention modules have to capture both global and local contexts, requiring a large capacity. That is not optimal compared with a specialized design. Taking the hardware design as an example, general-purpose hardware like CPUs is less efficient than specialized hardware like FPGAs. Here, we should specialize global and local contexts capture. When the model capacity is relatively large, the redundancy can be tolerated and may even provide better performance. However, when it comes to mobile applications, a model should be more efficient due to the computation and power constraints. Thus specialized contexts capture is more demanding. To tackle the problem, instead of having one module for "general" information, we propose a more specialized architecture, *Long-Short Range Attention (LSRA)*, that captures the global and local contexts separately.

As shown in Figure 3a, our LSRA module follows a two-branch design. The left branch captures global contexts, while the right branch models local contexts. Instead of feeding the whole input to both branches, we split it into two parts along the channel dimension, which will be mixed by the following FFN layer. Such practice reduces the overall computation by $2\times$. The left branch is a normal attention module as in Vaswani et al. (2017), while the channel dimension is reduced by half. For the right branch of local relationships, one natural idea is to apply convolution over the sequence. With a sliding window, the diagonal groups can be easily covered by the module. To further reduce the computation, we replace the normal convolution with a lighter version (Wu et al., 2019b) consisting of linear layers and depth-wise convolution. In this manner, we place the attention and the convolutional module side by side, encouraging them to have a different perspective of the sentence, globally and locally, so that the architecture can then benefit from the specialization and achieve better efficiency.

To have a better insight, we visualized the average attention weights of the same layer for a fully trained basic transformer and our Lite Transformer in Figure 3. It can be easily distinguished that instead of attempting to model both global and local contexts, the attention module in LSRA only focuses on the global contexts capture (no diagonal pattern), leaving the local contexts capture to the convolution branch.

## 5 EXPERIMENTAL SETUP

### 5.1 MOBILE SETTINGS

Most of machine translation architectures benefit from the large model size and computational complexity. However, edge devices, such as mobile phones and IoTs, are highly computationally limited. Those massive architectures are no more suitable for real-world mobile applications. To formalize the problem, we define the mobile settings for NLP models in terms of the amount of computation and the parameter numbers:

- The floating-point performance of the ARM Cortex-A72 mobile CPU is about 48G FLOPS (4 cores @1.5GHz). To achieve the peak performance of 50 sentences per second, the model should be less than 960M FLOPs (480M Mult-Adds). That is a common constraint in the computer vision community. For example, Liu et al. (2018) also uses 500M Mult-Adds as the constraint of its mobile setting. Therefore, we define the mobile settings for machine translation tasks: the computation constraint should be under **500M Mult-Adds** (or 1G FLOPs) with a sequence of 30 tokens (general length for machine translation).

- Additionally, we set a limitation for the parameters of the models. The constraint is based on the download and space limitation. Large mobile apps will take long time to be downloaded and even cost much money when using cellular networks. The run-time memory and disk size also constrain the parameter numbers. The parameters in MobileNet 7M parameters, we round it to the nearest magnitude, **10M parameters**, as our mobile constraint.

## 5.2 DATASETS AND EVALUATION

**Machine Translation.** The results are based on three machine translation benchmarks: For IWSLT'14 German-English (De-En), we follow the settings in Grave et al. (2017) with 160K training sentence pairs and 10K joint byte pair encoding (BPE) (Sennrich et al., 2016) vocabulary in lower case. For WMT English to German (En-De), we train the model on WMT'16 training data with 4.5M sentence pairs, validate on newstest2013, and test on newstest2014, the same as Wu et al. (2019b). Moreover, the vocabulary used a 32K joint source and target BPE. For WMT English to Franch (En-Fr), we replicate the setup in Gehring et al. (2017) with 36M training sentence pairs from WMT'14, validate on newstest2012 and 2013, and test on newstest2014. Also, the 40K vocabulary is based on a joint source and target BPE factorization.

For evaluation, we use the same beam decoding configuration used by Vaswani et al. (2017), where there is a beam size of 4 and a length penalty of 0.6. All BLEUs are calculated with case-sensitive tokenization[*], but for WMT En-De, we also use the compound splitting BLEU[†], the same as Vaswani et al. (2017). When testing, we average the last 10 model checkpoints for IWSLT De-En and take the model with the lowest perplexity on the validation set for the WMT datasets. We omit the word embedding lookup table from the model parameters since the number of entries in the table would highly differ for various tasks using transformer. For the Mult-Adds, we calculate the total number of multiplication-addition pairs for a model translating a sequence with the length of 30 to a sequence with the same length, which is the average length for sentence-level machine translation tasks.

**Abstractive Summarization.** We also evaluate our Lite Transformer on CNN-DailyMail dataset (Hermann et al., 2015) for abstractive summarization. The dataset contains 280K news articles paired with multi-sentence summaries. We truncate the articles to 1000 tokens and use a 30K BPE vocabulary. We use F1-Rouge as the metric, including Rouge-1 (R-1), Rouge-2 (R-2) and Rouge-L (R-L) (Lin, 2004)[‡]. We follow the generation settings in Lewis et al. (2019). We omit the word embedding lookup table and softmax layer from both the model parameters and #Mult-Adds calculation. #Mult-Adds is calculated for the documents with the input length of 30, 100, and 1000 and the output length of 60 (the average tokens for the output of CNN-DailyMail dataset).

**Language Modeling.** We test our Lite Transformer for language modeling task on WIKITEXT-103, which comprises about 100M tokens and a 260K BPE vocabulary. We evaluate the perplexity on both the validation set and the training set. The model parameters and #Mult-Adds are also computed for the input with a length of 30, 100, and 1000.

## 5.3 ARCHITECTURE

The model architecture is based on the sequence to sequence learning encoder-decoder (Sutskever et al., 2014). For machine translation, our baseline model is based on the one proposed by Vaswani et al. (2017) for WMT. For IWSLT, we follow the settings in Wu et al. (2019b). We also adopt the same model as on WMT for summarization task. For language modeling, our model is in line with Baevski & Auli (2019) but with smaller model dimension $d_{\text{model}} = 512$ and layer number

---

[*]https://github.com/moses-smt/mosesdecoder/blob/master/scripts/generic/multi-bleu.perl

[†]https://github.com/tensorflow/tensor2tensor/blob/master/tensor2tensor/utils/get_ende_bleu.sh

[‡]https://github.com/pltrdy/files2rouge

|  | #Parameters | #Mult-Adds | BLEU | ΔBLEU |
|---|---|---|---|---|
| Transformer (Vaswani et al., 2017) | 2.8M | 63M | 27.8 | – |
| LightConv (Wu et al., 2019b) | 2.5M | 52M | 28.5 | +0.7 |
| **Lite Transformer** (Ours) | 2.8M | 54M | **30.9** | **+3.1** |
| Transformer (Vaswani et al., 2017) | 5.7M | 139M | 31.3 | – |
| LightConv (Wu et al., 2019b) | 5.1M | 115M | 31.6 | +0.3 |
| **Lite Transformer** (Ours) | 5.4M | 119M | **32.9** | **+1.6** |
| Transformer (Vaswani et al., 2017) | 8.5M | 215M | 32.7 | – |
| LightConv (Wu et al., 2019b) | 8.4M | 204M | 32.9 | +0.2 |
| **Lite Transformer** (Ours) | 8.9M | 209M | **33.6** | **+0.9** |

Table 1: Results on IWSLT'14 De-En. Our Lite Transformer outperforms the transformer (Vaswani et al., 2017) and the Lightweight convolution network (Wu et al., 2019b) especially in mobile settings.

|  | #Parameters | #Mult-Adds | WMT'14 En-De | | WMT'14 En-Fr | |
|---|---|---|---|---|---|---|
|  |  |  | BLEU | ΔBLEU | BLEU | ΔBLEU |
| Transformer (Vaswani et al., 2017) | 2.8M | 87M | 21.3 | – | 33.6 | – |
| **Lite Transformer** (Ours) | 2.9M | 90M | **22.5** | **+1.2** | **35.3** | **+1.7** |
| Transformer (Vaswani et al., 2017) | 11.1M | 338M | 25.1 | – | 37.6 | – |
| **Lite Transformer** (Ours) | 11.7M | 360M | **25.6** | **+0.5** | **39.1** | **+1.5** |
| Transformer (Vaswani et al., 2017) | 17.3M | 527M | 26.1 | – | 38.4 | – |
| **Lite Transformer** (Ours) | 17.3M | 527M | **26.5** | **+0.4** | **39.6** | **+1.2** |

Table 2: Results on WMT'14 En-De and WMT'14 En-Fr. Our Lite Transformer improves the BLEU score over the transformer under similar Mult-Adds constraints.

$L = 12$ for the resource constraint. We use fairseq's reimplementation (Ott et al., 2019) of the transformer base model as the backbone.

In our architecture, we first flatten the bottleneck from the transformer base model and then replace the self-attention with the LSRA. More specifically, we use two specialized modules, an attention branch and a convolutional branch. Both the input and the output of the convolution are transformed by fully connected layers (GLU is applied for the input on WMT), and the kernel is dynamically calculated from the input using a fully connected layer in the WMT models. The kernel sizes are [3, 5, 7, 31×3] for both the encoder and the decoder (Wu et al., 2019b), and the number of heads for each module is 4 (half of the heads number in the transformer base model). The model for summarization is the same as the WMT model. For language modeling, the kernel sizes for the convolution branch are [15, 15, 31×4, 63×6].

## 5.4 TRAINING SETTINGS

All of our training settings for machine translation are in line with Wu et al. (2019b). We use a dropout of 0.3 for both the WMT and IWSLT datasets and linearly scale down the dropout ratio when shrinking the dimension of the embeddings for the WMT datasets. Same as Wu et al. (2019b), we apply Adam optimizer and a cosine learning rate schedule (Kingma & Ba, 2015; Loshchilov & Hutter, 2017) for the WMT models, where the learning rate is first linearly warm up from $10^{-7}$ to $10^{-3}$ followed by a cosine annealing with a single cycle. For IWSLT De-En, we use inverse square root learning rate scheduling (Vaswani et al., 2017) with the linear warm-up. We use the same training settings for summarization. For the language modeling task, the training settings are in line with Baevski & Auli (2019). We decrease the dropout ratio for the FFN layer by half in our Lite Transformer due to the flattened layer.

We train WMT and summarization models on 16 NVIDIA RTX 2080Ti GPUs and IWSLT De-En on a single GPU for 50K steps. We also accumulate the gradients for 8 batches before each model update (Ott et al., 2018). The gradients of IWSLT models are not accumulated. The maximum number of tokens in a batch is 4K for all the models. Label smooth of 0.1 is applied for the prior

| | #Params | #Mult-Adds | BLEU | GPU Hours | $CO_2e$ (lbs) | Cloud Computation Cost |
|---|---|---|---|---|---|---|
| Transformer (Vaswani et al., 2017) | 2.8M | 87M | 21.3 | 8×12 | 26 | $68 - $227 |
| Evolved Transformer (So et al., 2019) | 3.0M | 94M | 22.0 | 8×274K | 626K | $1.6M - $5.5M |
| **Lite Transformer** (Ours) | 2.9M | 90M | **22.5** | 8×14 | 32 | $83 - $278 |
| Transformer (Vaswani et al., 2017) | 11.1M | 338M | 25.1 | 8×16 | 36 | $93.9 - $315 |
| Evolved Transformer (So et al., 2019) | 11.8M | 364M | 25.4 | 8×274K | 626K | $1.6M - $5.5M |
| **Lite Transformer** (Ours) | 11.7M | 360M | **25.6** | 8×19 | 43 | $112 - $376 |

Table 3: Performance and training cost of an NMT model in terms of $CO_2$ emissions (lbs) and cloud compute cost (USD). The training cost estimation is adapted from Strubell et al. (2019). The training time for transformer and our Lite Transformer is measured on NVIDIA V100 GPU. The cloud computing cost is priced by AWS (lower price: spot instance; higher price: on-demand instance).

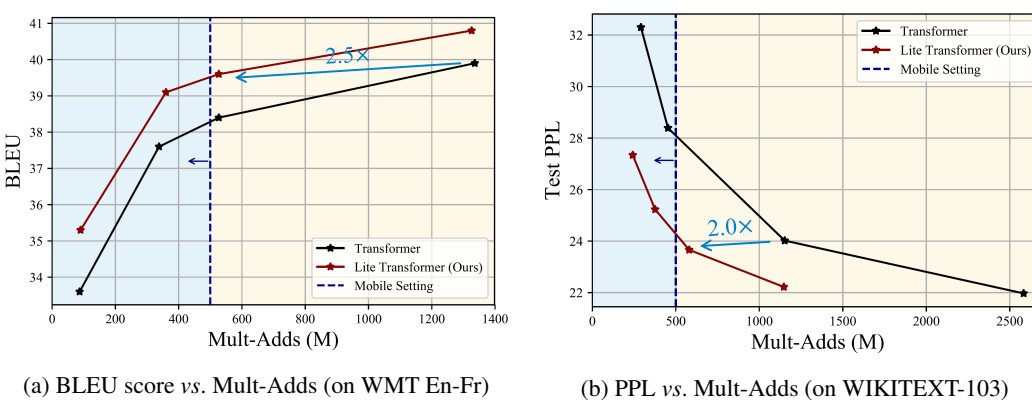

(a) BLEU score *vs*. Mult-Adds (on WMT En-Fr)          (b) PPL *vs*. Mult-Adds (on WIKITEXT-103)

Figure 4: Trade-off curve for machine learning on WMT En-Fr and language modeling on WIKITEXT-103 dataset. Both curves illustrate that our Lite Transformer outperform the basic transformer under the mobile settings (blue region).

distribution over the vocabulary (Szegedy et al., 2016; Pereyra et al., 2017). For language modeling, we train the models on 24 GPUs for 286K steps, the same as the settings in Baevski & Auli (2019).

# 6 RESULTS

## 6.1 MACHINE TRANSLATION

**Results on IWSLT.** We first report the results on IWSLT'14 De-En dataset. The baseline model is in line with Wu et al. (2019b), which provides the best results in the literature with 512 model dimension, 1024 FFN hidden dimension, and 4 heads for the attentions. Our Lite Transformer generally outperforms the transformer base under mobile constraints. With tighter computation limitations, our model achieves more significant improvement. That is because, when the dimension of the features decreases, it becomes much harder for the "general" attention to extract both the global and local features from the rather more compact information within the features. On the contrary, with the specialized LSRA, our model can capture the information from the features more efficiently.

In Table 1, we present the quantitative results of our Lite Transformer on IWSLT'14 De-En dataset, comparing to the transformer baseline as well as the LightConv (Wu et al., 2019b). Around 100M Mult-Adds, our model even achieves 1.6 BLEU score improvement than the transformer.

**Results on WMT.** We also show the result on the WMT'14 En-De and WMT'14 En-Fr dataset. Similar to the IWSLT, our Lite Transformer achieves a better trade-off with regard to transformer (Vaswani et al., 2017) against the total computation and the number of model parameters under mobile settings. The quantitative results in Table 2 indicates that our specialized Lite Transformer has 1.2 and 1.7 BLEU score improvement under 100M Mult-Adds and 0.5 and 1.5 around 300M Mult-Adds for

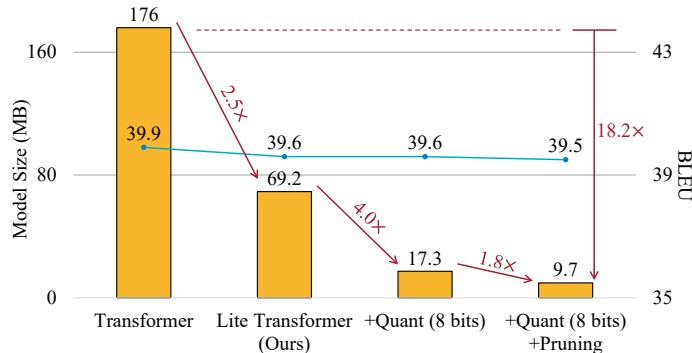

Figure 5: The model size and BLEU score on WMT En-Fr dataset with model compression. Our Lite Transformer can be combined with general compression techniques and achieves 18.2× model size compression. * 'Quant' indicates 'Quantization'.

|  | #Params | #MAdds (30) | #MAdds (100) | #MAdds (1000) | R-1 | R-2 | R-L |
|---|---|---|---|---|---|---|---|
| Transformer | 44.1M | 2.0G | 3.6G | 29.9G | 41.4 | 18.9 | 38.3 |
| **Lite Transformer** | **17.3M** | **0.8G** | **1.5G** | **12.5G** | 41.3 | 18.8 | 38.3 |

Table 4: Results on CNN-DailyMail dataset for abstractive summarization. Our Lite Transformer achieves similar F1-Rouge (R-1, R-2 and R-L) to the transformer (Vaswani et al., 2017) with more than 2.4× less computation and 2.5× less model size. "#MAdds (x)" indicates the #Mult-Adds required by the model with the input length of x.

|  | #Params | #MAdds (100) | #MAdds (1000) | Speed (tokens/s) | Valid ppl. | Test ppl. |
|---|---|---|---|---|---|---|
| Adaptive Inputs | 37.8M | 3.9G | 50.3G | 7.6K | 23.2 | 24.0 |
| **Lite Transformer** | 37.2M | 3.9G | 48.7G | 10.2K | **21.4** | **22.2** |

Table 5: Results on WIKITEXT-103 dataset for language modeling. We apply our Lite Transformer architecture on transformer base model with adaptive inputs (Baevski & Auli, 2019) and achieve 1.8 lower test perplexity under similar resource constraint.

WMT En-De dataset and WMT En-Fr dataset respectively. We also provide a tradeoff curve on WMT En-Fr in Figure 4a, where our Lite Transformer consistently outperforms the original transformer.

**Amenable to Compression.** As an efficient architecture, our Lite Transformer is orthogonal to general techniques for model compression (amenable to compression), *e.g.* pruning, and quantization. The results on WMT'14 En-Fr dataset with those techniques are shown in Figure 5. We quantize the model weight into 8 bits with K-means (Han et al., 2016) and prune the model according to the sensitivity of each layer (Han et al., 2015a). With the two model compression techniques, our method achieves 18.2× model size compression with negligible BLEU score degradation.

## 6.2 COMPARISON WITH AUTOMATED DESIGN

Comparing with the AutoML-based Evolved Transformer (ET) (So et al., 2019), our Lite Transformer also shows a significant improvement in mobile settings. Moreover, within mobile settings, the Lite Transformer outperforms the ET by 0.5 and 0.2 BLEU scores under 100M and 300M Mult-Adds, respectively, as shown in Table 3. Our architecture design is different from ET's design: ET stacks attentions and convolutions sequentially, while our Lite Transformer puts them in parallel; also, ET does not flatten the FFN.

Though nowadays, neural architecture search has been proved to be very powerful for searching in a large design space, the huge cost, more than 626155 lbs $CO_2$ emissions and more than 250 GPU years, cannot be ignored. Instead, careful human design with intuitions for specific tasks can also be a great choice in practice to save a large number of resources for the earth.

### 6.3 Abstractive Summarization and Language Modeling

We also test our Lite Transformer on longer input. In Table 4, we report results on CNN-DailyMail dataset for abstractive summarization. Our model achieves a similar F1-Rouge score as the transformer base model but requires $2.4\times$ less computation and $2.5\times$ storage resources. In Table 5, we provides the results of our Lite Transformer on WIKITTEXT-103 for language modeling task, compared with the adaptive inputs Baevski & Auli (2019) baseline. Under similar resource constraints, our Lite Transformer can achieve 3.9 and 1.8 lower perplexity on valid and test set, respectively. In Figure 4b, we show the tradeoff curve for our model and the baseline transformer model on WIKITEXT-103 between the test perplexity and the #Multi-Adds for input sentence with 30 tokens. It indicates that our Lite Transformer achieves consistent improvement over the original transformer, especially under mobile settings. Despite the translation tasks, the specialization design of LSRA is effective for larger scale language tasks.

## 7 Conclusion

In this paper, we presented *Long-Short Range Attention* (LSRA), where some heads specialize in the **local** context modeling while the others specialize in the **long-distance** relationship modeling. Based on this primitive, we design *Lite Transformer* that is specialized for the mobile setting (under 500M Mult-Adds) to facilitate the deployment on the edge devices. Our Lite Transformer demonstrates consistent improvement over the transformer on multiple language applications. It also surpasses the Evolved Transformer that requires costly architecture search under mobile settings.

**Acknowledgements.** We sincerely thank MIT-IBM Watson AI Lab, Facebook Faculty Award, Google-Daydream Research Award, and AWS Machine Learning Research Award for supporting this research.

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

## A.1 ADDITIONAL VISUALIZATION OF ATTENTION WEIGHTS

In this section, we show 3 more visualization of attention weights from both the base transformer and our LSRA. We use the smallest configuration in our paper for both models fully trained on WMT En-Fr translation and the attention weights are averaged among attention heads in the first layer. The sentences are sampled from this paper and the ICLR conference website.

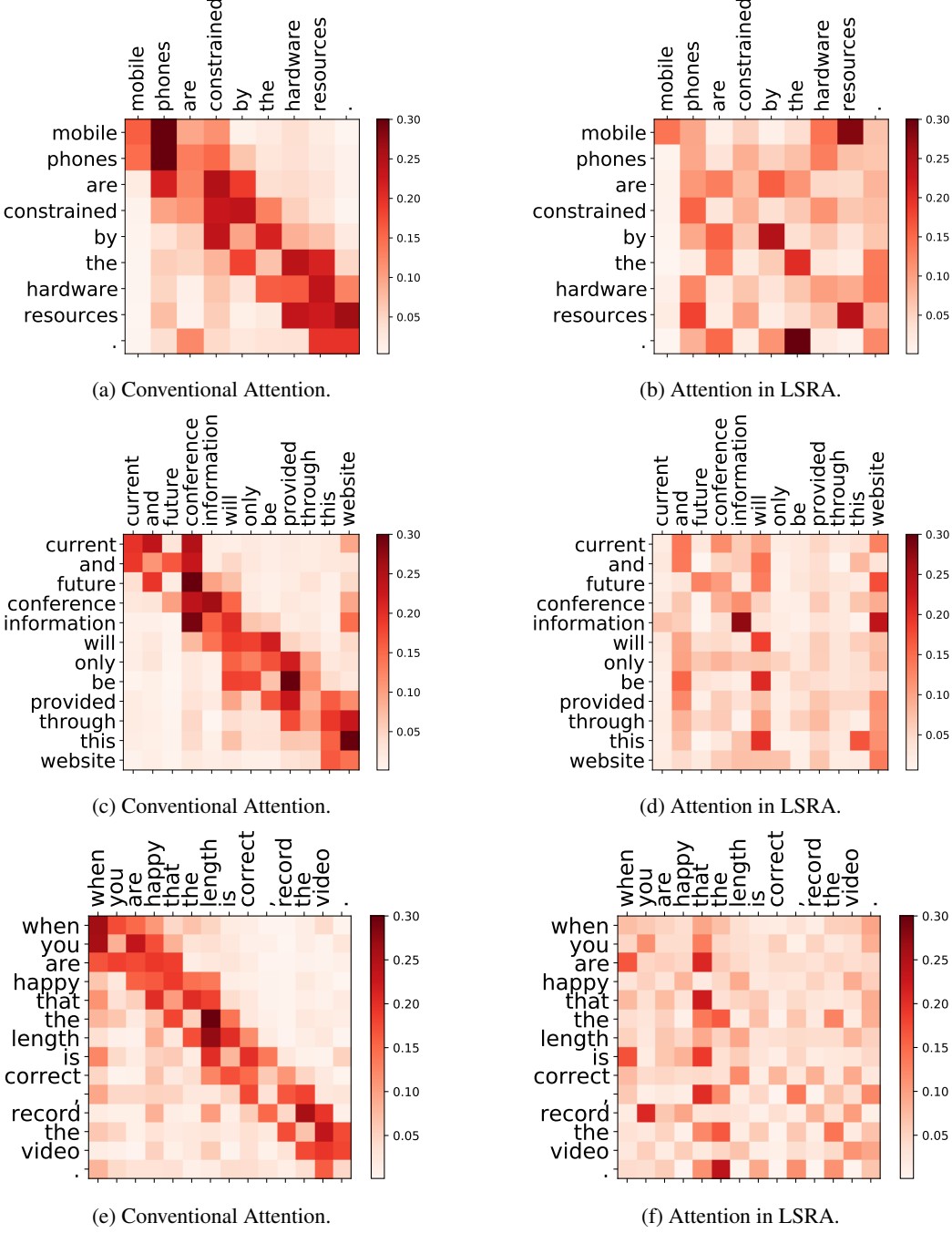

Figure A1: Conventional attention puts too much emphasis on local relationship modeling (see the diagonal structure). We specialize the local feature extraction by a convolutional branch which efficiently models locality so that the attention branch can specialize in global feature extraction (c). We provide some more visualizations in Section A.1.

