# OpenReview forum: "Lite Transformer with Long-Short Range Attention"
_ICLR.cc/2020/Conference — Accept (Poster)_

### Official Review · AnonReviewer2 · 2019-10-18
**Official Blind Review #2**

**Rating:** 6

**Review:**

This paper presents a new technique (LSRA) improving Transformer for constrained scenarios (e.g., mobile settings). It combines two attention modules to provide both global and local information separately for a translation task. In this manner, the authors place the attention and the convolutional module side by side, thus having different perspectives (globally and locally) of the sentence. They test their approach to 2 common translation benchmarks.

Enhancing deep learning model efficiency is very important, and the authors succeeded in reducing the computation costs and consequently, the CO2 emissions. But the evaluation results are not so impressive and go in line with other previous efficient deep learning approaches for different domains. I’m not an expert in NLP, but overall results of 10-1000x or more wall clock time reduction with <1-5% loss in accuracy are usually obtained for domains that have seen more optimization for mobile deployment (especially mobile-optimized CNNs like MobileNet). LSRA-based appraoch is slightly better than the original version of Transformer and its evolved version. From the latter (ET) authors seem to take the idea of parallel branches for their architecture. Also, adaptive attention span in Transformer models and all-attention layers have already been investigated to make networks more efficient and simpler for longer sentences. Include clearer ablation studies would be also interesting to support their findings and superior performance.

To summarize, the paper is addressing an important and interesting idea. It is, in general terms a nice engineering paper, but I am not sure about whether the developments and results are relevant/novel enough yet at this point to publish at ICLR.

-------------------------------------------------

Looking at the other comments and the feedback provided by the authors, I have a more positive feeling about the contributions of the paper which are now sufficiently demonstrated. Therefore, I increase my original recommendation to "Weak Accept".



**Experience Assessment:**

I have read many papers in this area.

**Review Assessment: Checking Correctness Of Derivations And Theory:**

N/A

**Review Assessment: Checking Correctness Of Experiments:**

N/A

**Review Assessment: Thoroughness In Paper Reading:**

N/A

---

> ### Author Response · Authors · 2019-11-13
> **Our Response to Reviewer 2**
>
> Thank you very much for your constructive comments.
>
> 1. Evaluation results
> We evaluated our Mobile Transformer on multiple machine translation datasets, including IWSLT De-En and WMT En-De. In our revision, we also added experimental results on WMT En-Fr (see Table 3 and Figure 4). Across all datasets, our Mobile Transformer consistently outperforms the vanilla Transformer and the AutoML-based Evolved Transformer. These are two strong baselines. Compared to the vanilla Transformer, we achieved twice the BLEU score improvement as the Evolved Transformer under similar constraints, which should be considered significant.
>
> On the other hand, our Mobile Transformer can reduce the computation by 2.5x with only 0.3 BLEU degradation and 15x with 5 BLEU degradation on the WMT En-Fr dataset. To achieve further compression, some general techniques (e.g., pruning, quantization) can be applied. In principle, we can safely quantize the model to 8 bits (4x) and prune the model by 50% (2x) without much loss of accuracy, which will give us around 120x reduction in model size.
>
> 2. Comparison with Evolved Transformer
> Our manual design is not the one presented in the ET’s paper even though it lies within its search space. This, we believe, is because NAS is limited by the sample efficiency and therefore cannot fully explore all the samples within its design space. We hope that our paper can raise people’s awareness about the importance of design insights: e.g., flatten the transformer to enlarge the attention’s computation, incorporate parallel branches that specialize in extracting features from different ranges.
>
> 3. Previous work
> We included the reference for adaptive span [1] and the all-attention [2] in our revision. Both methods are designed for the character-level language modeling, where the sequence is typically very long (>1000 tokens). This is drastically different from the machine translation, where the sequence is much shorter (<30 tokens). The adaptive span applies masks for the long-range relations, which will induce significant information loss when the sequence is relatively short. Also, these methods are orthogonal to our LSRA and can potentially be applied together.
>
> 4. Ablation study
> We included several ablation studies on the IWSLT dataset. We explored different combinations of two branches (attention+attention, convolution+convolution), and we also validated the effectiveness of flattening the FFN.
> ---------------------------------------------------------------------------------------------------
> | Model								|	#Mult-Adds	|	BLEU	|
> ---------------------------------------------------------------------------------------------------
> | Mobile Transformer (Ours)			|	209M		|	34.5		|
> | - with two branches of attention		|	232M		|	33.6		|
> | - with two branches of convolution	|	217M		|	33.8		|
> | - without flattening the FFN			|	207M		|	34.0		|
> ---------------------------------------------------------------------------------------------------
>
> We have also listed all other changes in our general response above. Please don’t hesitate to let us know for any additional comments on the paper.
>
> References:
> [1] Sainbayar Sukhbaatar, Edouard Grave, Piotr Bojanowski, and Armand Joulin, "Adaptive Attention Span in Transformers", ACL 2019.
> [2] Anonymous, "Augmenting Self-attention with Persistent Memory".

---

### Official Review · AnonReviewer3 · 2019-10-21
**Official Blind Review #3**

**Rating:** 8

**Review:**

The paper proposes Mobile Transformer, an efficient machine translation model, which achieves state-of-the-art results on IWSLT and WMT. The Mobile Transformer is base on long-short range attention (LSRA) modules that combine a depthwise convolution branch to encode the local information and a self-attention branch to capture the long-range information.

The main contribution of this paper includes
1. bottlenecks are not beneficial to 1D attention models
2. having both convolution and attention modules in parallel performs better and more efficient than having one of them alone. While LSRA is included in the search space of Evolved Transformer, surprisingly, their searching algorithm doesn't discover it. Evolved Transformer has either two convolution branches or two attention branches in parallel.

The paper is well written and easy to follow. The experiments are quite solid; however, it would be if the authors can report how Mobile Transformer performs on other language pairs or other NLP tasks.

Questions:
1. Do the attention maps in Figure 3 come from the average of multiple heads or just one of them?
2. The constraint for the mobile setting is set to 10M parameters. Can you justify why you choose it? In my opinion, memory footprint or inference time on mobile devices can be more realistic.
3. Regarding the design cost shown in Figure (b). Does the number for Mobile Transformer include the cost of all the experiments you ran to search for your Mobile Transformer?
4. I wonder if LSRA can also be applied to other tasks such as language modeling or reading comprehension.
5. In terms of inference latency, how much faster Mobile Transformer is compared to Transformer and LightConv?
6. Have you considered having the trade-off between having more parameters in the encoder or decoder?
7. Have you done any analysis on why all tokens in Figure (c) assign high weights to the <EOS> token?

**Experience Assessment:**

I have published one or two papers in this area.

**Review Assessment: Checking Correctness Of Derivations And Theory:**

N/A

**Review Assessment: Checking Correctness Of Experiments:**

I carefully checked the experiments.

**Review Assessment: Thoroughness In Paper Reading:**

I read the paper at least twice and used my best judgement in assessing the paper.

---

> ### Author Response · Authors · 2019-11-13
> **Our Response to Reviewer 3**
>
> Thank you very much for your encouraging and constructive comments.
>
> 1. Clarification of mobile settings
> We defined our mobile setting based on the real-world requirements for mobile applications and the conventions in the computer vision community. Please refer to our general response for more information.
>
> Instead of the memory footprint and inference time, we set our mobile setting based on the number of Mult-Adds and parameters due to the following reasons:
> (a) These two metrics are good indicators of the hardware resources required by the model: the number of Mult-Adds is correlated with energy consumption and inference time; the number of parameters indicates the storage on the mobile device.
> (b) The number of Mult-Adds and parameters are independent of the specific hardware (e.g., CPU, GPU), deep learning framework (e.g., PyTorch, TensorFlow), and backend acceleration (e.g., cuDNN, MKL-DNN).
>
> 2. Measured latency
> We measured the latency of our model and baselines on Raspberry Pi 4 (4 cores@1.5GHz):
> ---------------------------------------------------------------------------------
> |							|	Latency (ms/word)	|
> ---------------------------------------------------------------------------------
> | Transformer				|	95.0					|
> | LightConv					|	73.4					|
> | Mobile Transformer (Ours)	|	65.2					|
> ---------------------------------------------------------------------------------
> All of these models have the same BLEU score (34.1) on the IWSLT dataset. Our Mobile Transformer is 1.5x faster than the vanilla Transformer in terms of the measured latency.
>
> 3. Design cost
> We have run 5 experiments in total (on WMT En-De) to explore different implementations of convolution (i.e., vanilla convolution, depthwise convolution). However, we have not tuned our model architecture heavily: the two branches have the same embedding dimension and the same number of heads. Therefore, our total training cost (including tuning the model architecture) is around 16 * 5 = 80 GPU days, which is of the same magnitude as the training cost of the vanilla Transformer and is much lower than the search cost of the Evolved Transformer (about 250 GPU years).
>
> 4. Experiments on other language pair
> We also conducted experiments on one more language pair, WMT’14 English to French. Our Mobile Transformer consistently outperforms the Transformer by more than 1 BLEU score:
> ----------------------------------------------------------------------------------------------------------------------------------
> |							|	#Params	|	#Mult-Adds	|	BLEU	|	ΔBLEU	|
> ----------------------------------------------------------------------------------------------------------------------------------
> | Transformer				|	2.8M		|	87M		|	33.6		|	--		|
> | Mobile Transformer (Ours)	|	2.9M		|	90M		|	34.9		|	+1.3		|
> | Transformer				|	11.1M		|	338M		|	37.6		|	--		|
> | Mobile Transformer (Ours)	|	11.7M		|	360M		|	38.7		|	+1.1		|
> | Transformer				|	17.3M		|	527M		|	38.4		|	--		|
> | Mobile Transformer (Ours)	|	17.3M		|	527M		|	39.6		|	+1.2		|
> ----------------------------------------------------------------------------------------------------------------------------------
> Please refer to Table 3 and Figure 5 in the revised PDF for more information.
>
> 5. Attention maps
> In Figure 3, the visualization is based on the average attention maps of all heads in the same layer.
>
> 6. <EOS> token
> According to Clark et al. [1], each head of attention implicitly learns a function over the input sequence. As some head of attention only focuses on a subset of the sequence (e.g., nouns), the attention weights in the rows of all the other tokens (e.g., non-nouns) will be aggregated into some special entries (e.g., <EOS>). Thus, the attention weights on these special tokens will have a high value after taking the average.
>
> 7. Trade-offs between encoder and decoder
> We only experimented with the same number of layers in the encoder and decoder to have a fair comparison with the vanilla Transformer.
>
> 8. Other tasks
> Our LSRA is a general module that can, in principle, be plugged into the models for other tasks, including language modeling and abstractive summarization. This remains future work.
>
> We have also listed all other changes in our general response above. Please don’t hesitate to let us know for any additional comments on the paper.
>
> References:
> [1] Kevin Clark, Urvashi Khandelwal, Omer Levy, and Christopher Manning, "What Does BERT Look At? An Analysis of BERT’s Attention", BlackBoxNLP 2019.

---

### Official Review · AnonReviewer1 · 2019-10-25
**Official Blind Review #1**

**Rating:** 6

**Review:**



This paper claims to propose an extension of the Transformer architecture specialized for the mobile environment (under 500M Mult-Adds).
The authors propose their method called "Long-Short Range Attention (LSRA)," which separates the self-attention layers into two different purposes, where some heads focus on the local context modeling while the others capture the long-distance relationship.
They also demonstrate consistent improvement over the transformer on multiple datasets under the mobile setting.
It also surpasses the recently developed comparative method called "Evolved Transformer" that requires a far costly architecture search under the mobile setting.

This paper is basically well written and easy to follow what they have done.
The experimental results look good.

However, I have several concerns that I listed as follows.

1,
I am not sure whether my understanding is correct or not, but it seems that the proposed method, LSRA, is not a method specialized for mobile computation.
In fact, in the paper, they say, "To tackle the problem, instead of having one module for "general" information, we propose a more specialized architecture, Long-Short Range Attention (LSRA), that captures the global and local information separately."

There is no explicit discussion that LSRA is somehow tackling the mobile setting.
There is a large mismatch (gap) between the main claim and what they have done.
In other words, LSRA can be simply applied to standard-setting (non-mobile setting). Is there any reason that the proposed method cannot be applied to the standard-setting?
If my understanding is correct, the paper must be revised and appropriately reorganized to clear this gap.

2,
I am not convinced of the condition of the so-called "mobile setting (and also extremely efficient constraint)."
Please provide a clear justification for it.



**Experience Assessment:**

I have published in this field for several years.

**Review Assessment: Checking Correctness Of Derivations And Theory:**

I carefully checked the derivations and theory.

**Review Assessment: Checking Correctness Of Experiments:**

I carefully checked the experiments.

**Review Assessment: Thoroughness In Paper Reading:**

I read the paper thoroughly.

---

> ### Author Response · Authors · 2019-11-13
> **Our Response to Reviewer 1**
>
> Thank you very much for your constructive comments.
>
> 1. Specialization for mobile setting
> As mentioned in Section 4, the original multi-head self-attention captures ‘global’ and ‘local’ information in a single module. This unified design is not efficient especially when the computation budget is very limited (i.e., under the mobile setting). We need to make the module more specialized so that it can capture the context in a more efficient way. To this end, we proposed LSRA that captures the short-range information by convolution and the long-range information by attention. As shown in Figures 4 and 5, our LSRA achieves more improvements when the computation constraint is tighter. This is because the redundancy of the unified design is more severe under the mobile setting.
>
> We also conducted experiments under the standard setting on the WMT En-Fr dataset. Our Mobile Transformer still outperforms the vanilla Transformer under the standard setting:
> Base Transformer:                   40.0 BLEU @ 1336M Mult-Adds
> Mobile Transformer (Ours):   40.6 BLEU @ 1328M Mult-Adds
>
> 2. Clarification of mobile settings
> We defined our mobile setting based on the real-world requirements for mobile applications and the conventions in the computer vision community. Please refer to our general response for more information. We revised our paper accordingly in Section 5.
>
> We have also listed all other changes in our general response above. Please don’t hesitate to let us know for any additional comments on the paper.

---

### Author Response · Authors · 2019-11-13
**Our General Response**

We thank all reviewers for their comments. In addition to the specific response below, here we list the answers for some common questions, some additional experiments and the changes we made in the revision.

1. Clarifications of mobile setting
In this paper, we defined the mobile setting for NLP tasks, which is under 500M Mult-Adds and 10M parameters. These two constraints are based on the real-world requirements for mobile devices and the conventions in the computer vision community.
(a) Mult-Add constraints: The floating-point performance of the ARM Cortex-A72 mobile CPU is about 48G FLOPS (4 cores @1.5GHz). To achieve the peak performance of 50 sentences per second, the model should be less than 960M FLOPs (480M Mult-Adds). This is a common constraint in the computer vision community. For example, MobileNet [1, 2] is less than 500M Mult-Adds; PNAS [3] uses 500M Mult-Adds as the constraint of its mobile setting.
(b) Parameter constraints: The constraint for the parameters is based on the download limitation. When an application is larger than 100MB, it cannot be downloaded with 4G LTE (only with WIFI) in the App Store. Therefore, the number of parameters for a mobile model should be limited. As MobileNet V2 (1.4) contains around 7M parameters, we round it to the nearest magnitude, 10M parameters, as our constraint.

2. Additional experiments
(a) We conducted experiments on one additional language pair, WMT’14 English to French. Our Mobile Transformer outperforms the vanilla Transformer by more than 1 BLEU score.
(b) We evaluated our Mobile Transformer on the standard setting. It outperforms the vanilla Transformer by 0.6 BLEU score on WMT’14 English to French.
(c) We conducted some additional ablation studies including different combinations of two branches and different numbers of heads in LSRA.

3. Other changes
(a) We included an additional comparison with the Evolved Transformer under the 360M Mult-Adds constraints, where the Evolved Transformer achieves 25.4 BLEU @ 364M Mult-Adds, and our Mobile Transformer achieves 25.8 BLEU @ 360M Mult-Adds.
(b) We removed the extremely efficient computation constraint for clarity.
(c) We added more discussions with some related papers.

References:
[1] Andrew Howard, Menglong Zhu, Bo Chen, Dmitry Kalenichenko, Weijun Wang, Tobias Weyand, Marco Andreetto, and Hartwig Adam, "MobileNets: Efficient Convolutional Neural Networks for Mobile Vision Applications", arXiv 2017.
[2] Mark Sandler, Andrew Howard, Menglong Zhu, Andrey Zhmoginov, and Liang-Chieh Chen, "MobileNetV2: Inverted Residuals and Linear Bottlenecks", CVPR 2018.
[3] Chenxi Liu, Barret Zoph, Maxim Neumann, Jonathon Shlens, Wei Hua, Li-Jia Li, Li Fei-Fei, Alan Yuille, Jonathan Huang, and Kevin Murphy, "Progressive Neural Architecture Search", ECCV 2018.

---

### Decision · Program_Chairs · 2019-12-19

**Decision:**

Accept (Poster)

**Comment:**

This paper presents an efficient architecture of Transformer to facilitate implementations on mobile settings. The core idea is to decompose the self-attention layers to focus on local and global information separately. In the experiments on machine translation, it is shown to outperform baseline Transformer as well as the Evolved Transformer obtained by a costly architecture search.
While all reviewers admitted the practical impact of the results in terms of engineering, the main concerns in the initial paper were the clarification of the mobile settings and scientific contributions. Through the discussion, reviewers are fairly satisfied with the authors’ response and are now all positive to the acceptance. Although we are still curious how it works on other tasks (as the title says “mobile applications”), I think the paper provides enough insights valuable to the community, so I’d like to recommend acceptance.